# Hybrid Nanomaterial of Graphene Oxide Quantum Dots with Multi-Walled Carbon Nanotubes for Simultaneous Voltammetric Determination of Four DNA Bases

**DOI:** 10.3390/nano13091509

**Published:** 2023-04-29

**Authors:** Qusai Hassan, Chevon Riley, Meissam Noroozifar, Kagan Kerman

**Affiliations:** 1Department of Physical and Environmental Sciences, University of Toronto Scarborough, 1265 Military Trail, Toronto, ON M1C 1A4, Canada; qusai.hassan@mail.utoronto.ca (Q.H.); chevon.riley@mail.utoronto.ca (C.R.); 2Department of Chemistry, University of Toronto, 80 St. George Street, Toronto, ON M5S 3H6, Canada

**Keywords:** electrochemical sensor, graphene oxide quantum dots, multi-walled carbon nanotubes, hybrid nanomaterial, simultaneous determination, DNA

## Abstract

In this proof-of-concept study, a novel hybrid nanomaterial-based electrochemical sensor was developed for the simultaneous detection of four DNA bases. For the modification of the working electrode surface, graphene oxide quantum dots (GOQDs) were synthesized using a solvothermal method. GOQDs were then used for the preparation of a hybrid nanomaterial with multi-walled carbon nanotubes (GOQD-MWCNT) using a solvothermal technique for the first time. Transmission electron microscopy (TEM) was used to characterize the GOQDs-MWCNTs. A glassy carbon electrode (GCE) was modified with the GOQDs-MWCNTs using Nafion™ to prepare a GOQD-MWCNT/GCE for the simultaneous determination of four DNA bases in phosphate buffer solution (PBS, pH 7.0) using differential pulse voltammetry (DPV). The calibration plots were linear up to 50, 50, 500, and 500 µM with a limit of detection at 0.44, 0.2, 1.6, and 5.6 µM for guanine (G), adenine (A), thymine (T) and cytosine (C), respectively. The hybrid-modified sensor was used for the determination of G, A, T, and C spiked in the artificial saliva samples with the recovery values ranging from 95.9 to 106.8%. This novel hybrid-modified electrochemical sensor provides a promising platform for the future development of a device for cost-effective and efficient simultaneous detection of DNA bases in real biological and environmental samples.

## 1. Introduction

The key to the identity of any living organism lies in its genetic code, which is comprised of deoxyribonucleic acid (DNA). DNA holds the instructions for the growth, development, reproduction, and functioning of all organisms. Although the genetic instructions from DNA result in the formation of complex protein structures, DNA can ultimately be broken down into one of the four nitrogen derivative nucleobases (adenine (A), guanine (G), cytosine (C), and thymine (T)). These four nucleobases subsequently fall into two subcategories: purines (paired together with two hydrogen bonds (T and A)) and pyrimidines (paired together with three hydrogen bonds (G and C) [1]. DNA naturally undergoes mutations due to environmental factors, aging, and lifestyle behaviours such as diet induced biochemical alteration and smoking [2]. Common mutations are mismatched nitrogen derivative nucleobases due to erroneous duplication of nucleosides that are undamaged, adducts that form on DNA or breakage of single and double strands [2]. With mutations in DNA, the range of genetic diseases is endless, such as cancer [3,4]. There has yet to be a cure for these genetically inherited diseases; thus, sensitive detection of the respective nucleobases may assist in the development of early anti-cancer agents that target specific DNA bases such as cis-platin. Furthermore, several metabolic diseases resulting in abnormal levels of DNA bases such as HIV/AIDS, gout, Lesch–Nyhan disease, and prostatitis can be diagnosed earlier [5,6].

To date, there are a wide variety of analytical methods that can detect DNA. One such method is ultra-high-performance liquid chromatography enabled with electrospray ionization and tandem mass spectrometry. These methods were used to simultaneously detect 13 nucleosides and nucleobases in Cordyceps sinensis [7]. Capillary electrophoresis was used to detect A and C with high sensitivity [8,9,10]. Chemical luminescence studies made molecular recognition and DNA detection simple, sensitive and easy [10]. While Raman spectroscopy aided nucleotide identification and DNA sequencing [11]. Gas chromatography, spectrophotometric methods and high-performance liquid chromatography detected DNA bases in several different sample matrices [12,13,14]. These sophisticated analytical methods provided excellent quantification results; however, they may not be cost- and time-effective as the operations are complex and time consuming, requiring skilled technicians, making the cost for running these instruments expensive for routine analyses.

With the huge commercial success of glucose biosensors, electrochemistry has gained attention for the analysis of other biomolecules with high sensitivity [15]. Furthermore, due to the simplicity of executing operations, portable devices have been developed to support electrochemical analysis at point-of-care sites [15]. As a result of non-specific adsorption issues, biosensors are generally challenged with the adsorption of the oxidized DNA products on the surface of the electrode. Thus, researchers have been attempting to address the issue of non-specific adsorption [16]. Oliveira-Brett et al. [17] reported the electrochemical oxidation of both pyrimidine and purine compounds using a glassy carbon electrode (GCE) in basic conditions. A linear range of detection (from 1 to 20 µM for pyrimidines and from 0.2 to 10 µM for purines) was established for the four nucleobases. GCE implements both the ceramic and glassy properties of graphite, which include hardness, low electrical resistance, extreme resistance to chemical contamination, low density and friction, impermeability to gases and liquids, high-temperature resistance, and low thermal resistance, making GCE popular in electrochemistry. Suprun et al. reported the electrochemical behavior of DNA bases on carbon electrodes [18]. Arul et al. used a modified GCE with AgNPs embedded covalent organic framework for the simultaneous determination of DNA bases [19]. Carbon nanotubes (CNTs) pose great chemical stability. When used for the modification of sensor surfaces, they aid in promoting electron transfer. Since 1991, the mechanical properties and electrical conductivity of CNTs have been intensively studied [20]. To date, CNTs can be found in the forms of either multi-walled carbon nanotubes (MWCNTs) or single-walled carbon nanotubes (SWCNTs). MWCNTs contain rolled-up graphene tubules that are centric to each other. SWCNTs, on the other hand, possess only one graphite sheet rolled up [21]. In order to prepare a MWCNT-modified graphite paste electrode (MWCNT-GPE), MWCNTs are usually mixed with a graphite paste that contains mineral oil, bromoform, Nafion™, or liquid paraffin, and then packed into a glass or Teflon tube with a copper wire for electrical contact [22]. These electrodes could then be utilized to probe electrocatalytic and bioelectrochemical reactions [22,23,24]. CNTs are also widely used in the development of sensing surfaces in field-effect transistors and tips in scanning probe microscopy [25,26].

The latest research frontiers in carbon-based nanomaterials, carbon dots (CDs), graphene quantum dots (GQDs) and graphene oxide quantum dots (GOQDs) have opened new horizons in the field of electrochemical sensors. These nanomaterials present an excellent opportunity for biological sensing as they have great biocompatibility and intrinsically low toxicity [27]. They possess abundant edge sites for functionalization making them versatile as they present different possibilities for modification with attractive surface chemistries and other nanocomposites/nanomaterials. Furthermore, they maintain excellent electronic properties and large electro-active surface areas while being highly soluble in many solvents and relatively inexpensive to produce [28].

GOQDs and CDs can be used as electrode surface nanocomposites or signal tags for developing electrochemical bio-sensing strategies [29]. Due to its size, GOQDs have more available edges, surface active sites, and a larger specific surface area than CDs. However, due to the long synthesis process of GOQDs, there is minimal commercial availability. Nevertheless, GOQDs as zero-dimensional graphene can be a great candidate for optical and electrochemical sensing applications [28,30].

In our previous studies, we modified GCEs with GO/MWCNTs [5] and GO nanoribbons in chitosan [31] for the simultaneous determination of four DNA bases. We have also shown that incorporating sugarcane biochar along with methylene blue in a graphite paste electrode can enhance the sensing of DNA bases using biocompatible material [6]. Researchers have also coupled carbon fiber microelectrodes with fast-scan cyclic voltammetry to detect DNA bases [32]. Other reports included the use of epitaxial graphene to detect DNA bases [33]. Herein, GOQDs were synthesized and utilized for the development of a novel MWCNT/GOQDS hybrid nanomaterial via a solvothermal technique. The hybrid nanomaterial was incorporated into Nafion™ polymer and drop-cast onto a GCE. Differential pulse voltammetry (DPV) was employed in phosphate buffer solution (PBS, pH 7.0) for the simultaneous determination of G, A, C, and T. After the optimization of determination conditions, the hybrid nanomaterial-modified sensor was used for the determination of G, A, C, and T in artificial saliva using the standard addition method. This nanomaterial presents an opportunity to develop inexpensive, biocompatible materials that can be made into in vivo sensors that can be easily functionalized based on the analyte being sensed [27]. Ultimately, this sensor can be developed into a point-of-care device that can monitor, in real time, diseases that result in DNA nucleobase imbalances such as HIV/AIDS, gout, Lesch–Nyhan disease, and prostatitis [5,6].

## 2. Materials and Methods

### 2.1. Materials and Reagents

Guanine (G), adenine (A), thymine (T), cytosine (C), Nafion™, NaNO3, NaOH, potassium ferricyanide(III) (K_3_[Fe(CN)_6_]), potassium ferrocyanide(II) trihydrate (K_4_[Fe(CN)_6_]·3H_2_O), potassium permanganate (KMnO₄), acetic acid, HCl, and H_2_SO_4_ were all purchased from Sigma-Aldrich (Oakville, ON, Canada). Phosphate electrolytes (from pH 3.0 to 6.0) and phosphate buffer solutions (PBS, from pH 6.0 to 8.0) of 0.2 M phosphoric acid (H_3_PO_4_) (Fischer Scientific, Mississauga, ON, Canada) were prepared according to the previous protocols using a solution of concentrated NaOH to adjust the pH [5,31]. The pH for each solution was measured prior to conducting the experiment. Multi-walled carbon nanotubes (MWCNTs, 13–18 nm) were purchased from Cheap Tubes Inc. (Cambridgeport, VT, USA). For the cleaning of GCE surfaces, various sizes of alumina powder at 1.0 µm, 0.3 µm, and 0.05 µm were obtained from CH Instrumental Inc. (Austin, TX, USA). Solutions of the aforementioned four DNA bases were prepared fresh by dissolving the compounds in a 40:1 deionized water:concentrated NaOH solution which was sonicated for 5 min. The final concentrations of DNA bases were 0.01 M for G and A and 0.05 M for T and C. All other chemicals and reagents were purchased from Sigma-Aldrich (Oakville, ON, Canada) and used without further purification. All stock analyte solutions were prepared using the sterile 18.2 µΩ ultra-pure water obtained from a Cascada LS water purification system, which is equipped with a UV-lamp and a 0.2 μm bacterial filter (Pall Co., Mississauga, ON, Canada).

### 2.2. Instrumentation

All electrochemical measurements were performed at room temperature using Autolab Potentiostat/Galvanostat (PGSTAT 302N, Metrohm AG, Herisau, Switzerland) controlled by NOVA™ 2.1.2 (Metrohm AG, Herisau, Switzerland) software. The electrochemical cell was composed of the GCE as the working electrode (diameter 2 mm), a reference electrode (saturated Ag/AgCl, 3 M KCl), and a platinum wire as the counter electrode. Cyclic voltammograms were run at bare GCE in a solution containing 5 mM of both K_3_[Fe(CN)_6_] and K_4_[Fe(CN)_6_] with 100 mM of KCl in 0.2 M PBS solution (pH 7.0) at a scan rate of 100 mV/s in order to estimate the formal potential which was found to be 0.23 V (Appendix A) [34]. Differential pulse voltammetry (DPV) was conducted in a potential window from 0 V to +1.5 V at a step potential of 5 mV, a modulation amplitude of 25 mV and modulation time of 0.05 s with an interval time of 0.5 s. All voltammograms were processed by smoothing and baseline correction with a moving average of window size 1 using NOVATM 2.1.2 software. X-ray photoelectron spectroscopy (XPS) was performed using a Thermo Scientific K-Alpha spectrometer (Mississauga, ON, Canada) equipped with a monochromated Al Kα X-ray source (1486.6 eV), using an acquisition angle of 90° with a 20 eV pass energy, and the acquisition chamber at a pressure of 10–8 mbar (Appendix A). A VWR SB70P pH meter (Thermo-Fisher, Mississauga, ON, Canada) was used to measure the pH of the electrolyte solutions prior to use for each experiment. All mixtures were sonicated using a VWR B2500A-DTH ultra-sonicator (Thermo-Fisher, Mississauga, ON, Canada). The transmission electron microscopy (TEM) images were collected by a Hitachi H-7500 transmission electron microscope (Hitachi, Chiyoda, Tokyo, Japan). Atomic force microscopy (AFM) of the GOQD was performed using NanoBrook Omni (Brookhaven Instruments, Holtsville, NY, USA). AFM images were taken in the Quantitative Imaging (QI) mode using a SNL-10A tip. The spring constant of the cantilever was equal to 0.35 N m^−^^1^ with a 2 nm tip radius, and the setpoint for imaging was 2 nN.

### 2.3. Synthesis of GOQDs

First, GO was prepared from graphite using a modified Hummer’s method [35] as shown in Figure 1. Briefly, 3 g of NaNO_3_ was dissolved in 150 mL of concentrated sulfuric acid. Subsequently, 3 g of graphite was added to the mixture and stirred. A desired amount (18 g) of KMnO_4_ was added to the mixture and stirred for 4 h at 10 °C. As the mixture was stirred, the colour changed from purple to brown, indicating the oxidation of graphite. Once the stirring was completed, 300 mL of deionized water was added to stop the reaction. The mixture was then centrifuged at 10,000 rpm for 15 min to collect the product. The product was washed and centrifuged 3 times with ultra-pure water and dried at 70 °C overnight. The synthesis of GOQDs was adapted from Liu et al. [36]. Briefly, 500 mg of the GO was mixed with 5 g potassium permanganate and then 10 mL of concentrated sulfuric acid step by step under magnetic stirring. The mixture was transferred to an ice bath and then 5 mL of concentrated nitric acid was added slowly. The mixture was transferred to a 25 mL Teflon-lined stainless-steel autoclave container and stirred. The final mixture was kept at 180 °C overnight in the oven (Thermo-Fisher, Mississauga, ON, Canada). The final product was transferred to an ice bath and 50 mL of ultra-pure water was added. The final product was separated from a solution with a centrifuge (12,000 rpm for 1 h, Millipore-Sigma, Oakville, ON, Canada) and washed with ultra-pure water three times and separated. Finally, GOQDs were dried overnight at 70 °C.

### 2.4. Synthesis of GOQD-MWCNT Hybrid Nanomaterial

The GOQD-MWCNT hybrid nanomaterial was prepared (Figure 1) by mixing 10 mg of GOQDs and 50 mg MWCNTs with 5 mL of ultra-pure water in a scintillation vial. The mixture was placed in an ultrasonic bath for 15 min at 40 °C, which was then transferred to a Teflon-lined stainless-steel autoclave container and kept at 180 °C for 24 h. The resulting compound was washed with ultra-pure water three times and separated from a solution with a centrifuge (12,000 rpm for 1 h). Finally, the GOQD/MWCNT hybrid nanomaterial was dried overnight at 70 °C.

### 2.5. Modification of GCE Surface with GOQD-MWCNT

GCEs (2 mm in diameter) were polished using alumina powder paste of differing sizes (1.0, 0.3 to 0.05 µm) on a Nylon polishing pad until the mirror was shiny. The GCEs were then sonicated in deionized water followed by ethanol for 5 min to remove any adhering alumina. The electrodes were then electrochemically cleaned using CV (15 cycles) in 1.0 M H_2_SO_4_ solution in a potential window of −1.5 V to 1.5 V at a scan rate of 100 mV/s (Appendix A). Then, 2 mg of this GOQD-MWCNT hybrid nanomaterial was dispersed in 20 µL of Nafion™ (1.5% *v*/*v*). Next, 1 µL of the prepared GOQD- MWCNT in Nafion™ was drop-cast onto the surface of the activated GCE and dried under heat for 30 min to form a stable layer on the GCE surface. Before each measurement, the nanocomposite-modified GCEs were electrochemically equilibrated by running 30 cycles of CV in 0.2 M PBS (pH 7.0) from −0.8 V to 1.6 V at 100 mV/s.

### 2.6. Artificial Saliva Sample Preparation

For the preparation of artificial saliva samples, a method described by Madsen et al. [37] was followed. An aliquot (25 mL) of artificial saliva solution (2.5 mM NaHCO_3_, 7.4 mM NaCl, 10 mM KCl, 2 mM CaCl_2_·2H_2_O, 6.4 mM NaH_2_PO_4_·2H_2_O in deionized water) was prepared. An aliquot (400 μL) of the artificial saliva was then diluted with 1600 μL of 10 mM PBS solution (pH 7.0). The four DNA base solutions were then used to spike the diluted samples using the standard addition method.

## 3. Results and Discussion

### 3.1. TEM and AFM Characterization

TEM was used for the characterization of the GOQD, the MWCNTs and the GOQDs-MWCNT hybrid nanomaterial. The samples for TEM images were prepared by dispersing the nanocomposite in iso- propanol and an aliquot (5 µL) of the suspension was put on the carbon film with 300 mesh and copper grid (CF300-CU). The results are shown in Figure 1. Figure 1A,B show the TEM of MWCNTs with different magnifications. Figure 1E,F show the TEM of GOQD-MWCNT with different magnifications. In comparison with the TEM images of MWCNTs alone, GOQD-MWCNT images displayed the attached clusters of GOQDs, such as small nanoparticles on the surface of MWCNTs as indicated by arrows. Based on the TEM images, we estimated that the sizes of the GOQDs were approximately 15–20 nm in diameter, which agreed with previous literature [29,38,39]. For comparison, the TEM image of GOQDs is shown in Figure 1C. Figure 1D shows the AFM characterization of the GOQD, which was prepared by dispersing the GOQD in isopropanol, then adding a 10 μL aliquot onto a glass slide and allowing it to dry. The lateral sizes of GOQDs are in the range of 25–45 nm. The AFM image shows consistent sizing with the TEM images.

### 3.2. Comparison Study

The electrochemical properties of the GOQD-MWCNT/GCE were investigated using DPV in 0.2 M PBS (pH 7.0). Figure 2 shows the typical DPV curves of bare GCE, MWCNT/GCE and GOQD- MWCNT/GCE in the presence of a mixture of G, A, T and C. The GOQD-MWCNT/GCE showed a significant increase in the peak currents of G, A, T and C, in comparison with those obtained using a bare GCE and MWCNT/GCE. Thus, this was clear evidence for the high electrocatalytic activity of GOQD-MWCNT in the electrochemical determination of G, A, T and C. The enhanced electrocatalytic activity of GOQD-MWCNT could be ascribed to the synergistic effect of the GOQDs with the MWCNTs. Due to the unique properties of GOQDs, the synergistic effect depends on the shape, size and high surface area of immobilized GOQDs on the surface of MWCNTs [40].

### 3.3. pH Effect

Using the GOQD-MWCNT/GCE, the effect of the pH on the simultaneous determination of the four DNA bases was studied. Figure 3A shows the DPVs of the DNA bases spiked (16 µM G, A; 160 µM T, C) under various pH conditions ranging from pH 3.0 to 8.0. Based on this Figure, as the pH increased, the peak potentials of the analytes shifted to more negative values. This shift is caused by protons’ involvement in the analytes’ electrooxidation mechanism, which results in changes in their oxidation peak potentials [5,31,41,42,43,44,45]. This highlights the importance of pH control in electrochemical measurements, particularly when detecting proton-involved analytes. Figure 3B demonstrates the linear relationship between the oxidation potential (Ep) and pH for the four DNA bases. The slope of the linear relationship indicated the relative number of protons and electrons that were exchanged in the oxidation reaction of the DNA bases [5,31,41]. The equations for these relationships and pH of the four DNA bases are shown in Figure 3B. Based on the equations, the observed slopes for G, A, T and C were 0.0562, 0.0660, 0.0597, and 0.0763 V/pH, respectively. These values were compatible with value 𝑚 of the Nernstian slope (0.059 𝑛mV/pH [42,43,44,45], where m and n are the numbers of protons and electrons exchanged, respectively). The results indicated a proton-coupled redox process with an equivalent number of protons and electrons being exchanged at GOQD-MWCNT/GCE.

Based on these results, the proposed electrooxidation mechanisms for the four DNA bases are shown in Figure 2, Equations (1)–(4), which are in agreement with the previously reported results in the literature [5,31,46,47,48,49]. Additionally, based on Figure 3A, the voltammogram displayed well-defined peaks pH 7.0 with high current intensities for G, A, T and C at low oxidation potentials (except pH 8.0) compared with the results obtained using other pH conditions. In addition, pH 7.0 was close to the physiological pH of 7.4, thus, pH 7.0 was chosen as the optimum pH condition for the simultaneous determination of four DNA bases using GOQD-MWCNT/GCE.

### 3.4. Interference Study

Four DNA bases can coexist at varying degrees in different real-life samples such as urine, sperm DNA, serum, and saliva [5,32,50,51]. Furthermore, other metabolites may coexist in these real samples, which may interfere with the detection of these DNA bases such as glucose, ascorbic acid, and cysteine [50]. Hence, the study of the standard addition of each DNA base is important to see whether the biomolecule can disrupt the detection of the other species. In all experiments conducted, the concentration of one species was increased while the others were kept at a high constant concentration in the solution throughout to test the electrooxidation processes of each analyte. This standard addition study for G, A, T and C is shown in Figure 4A–D. Based on these results, the peak current of one analyte increased in a concentration-dependent manner while the peak current signals of the other three bases remained unaffected and constant with a relative standard deviation of ±5.0% (*n* = 3). We concluded that the oxidation processes of four DNA bases took place without interfering with each other using GOQD-MWCNT/GCE. Additionally, an interference study was conducted using varying concentrations of ascorbic acid and glucose to determine whether these species interfered with the detection of the four DNA bases (Appendix A). As shown in Appendix A, even in the cases where either ascorbic acid or glucose were in excess of the other DNA bases, negligible interference was observed for the detection of the four DNA bases.

### 3.5. Calibration Study

DPV measurements were performed to explore the relationship between the peak currents and concentrations of G, A, T and C. The concentrations of four analytes were increased in a stepwise manner, and the measurements were stopped when the saturation of peak current signals was reached. The voltammograms are shown in Figure 5A with the corresponding calibration curves and equations in Figure 5B–E. For each analyte, the peak currents were subtracted by the background current of GOQD- MWCNT/GCE to produce ΔI, and ΔI was plotted against the concentration of the analytes. All four calibration curves showed one segment. LOD was calculated by using the equation 3𝑆*D*/*m* where *SD* is the standard deviation of the blank voltammograms (*n* = 10) and *m* is the slope of the curve obtained from the calibration curve. In addition, the repeatability of the GOQD/MWCNT/GCE was investigated using the DPV technique for 10 consecutive trials while simultaneously detecting G, A, T, and C. The results in Figure 6 show that the relative standard deviations for G, A, T, and C were 2.36, 2.02, 2.31, and 1.58%, respectively. Furthermore, the GOQD/MWCNT/GCE stability was investigated by measuring DPV signals for simultaneous detection of G, A, T, and C for 6 months in 0.2 M phosphate electrolyte solution (pH 7.0). The same concentration of DNA bases was used to record the DPV signals. The retained peak potential and peak current (I_pa_) values for G, A, T, and C were 98.3, 97.6, 98.1, and 97.5%, highlighting the excellent stability of the PBNS-PANI/GPEs. These findings support the GOQD/MWCNT/GCE’s excellent stability and repeatability in the simultaneous determination of G, A, T, and C.

Table 1 summarizes the analytical performance of the modified electrode in comparison with similar sensors reported in recent literature. As shown in this table, three of the recently reported modified electrodes were used for the simultaneous determination of only G and A. We observed that the GOQD-MWCNT/GCE could provide satisfactory linear range and LOD in comparison with the other modified electrodes.

### 3.6. Artificial Saliva Sample Analyses

Artificial saliva solutions were used for the preparation of the spiked solutions that included four DNA bases. The standard addition method was used to quantify G, A, T and C in three separate trials. The results are shown in Table 2 with a good recovery value which ranged from 95.9 to 106.8% (n = 3). The good recovery values obtained were a promising indication of minimal interference from interfering molecules of various salts present in artificial saliva. These results demonstrated that the GOQD-MWCNT/GCE had good potential for the analyses of DNA bases in real biological fluids.

## 4. Conclusions

In this proof-of-concept work, the GOQD-MWCNT hybrid nanomaterial was developed and cast on the GCE surface using Nafion™. To the best of our knowledge, this hybrid nanomaterial was used as an electrochemical sensor surface for the first time. The modified electrode was used for the simultaneous oxidation of four DNA bases: G, A, T, and C. The calibration curves were linear up to 50.0 µM for both G and A and up to 500.0 µM for e T and C. The LODs were 0.44, 0.2, 1.6, and 5.6 µM for G, A, T, and C, respectively. Finally, the GOQD-MWCNT/GCE was used for the simultaneous determination of spiked G, A, T, and C in the artificial saliva samples with the recovery values ranging from 95.9 to 106.8%. We envisage that the electrochemical sensor provides a promising platform for the development of a rapid prescreening tool for the simultaneous detection of DNA bases in clinical diagnostic studies as well as in the quality control assessment departments of the pharmaceutical and food industries.

## Data Availability

The data presented in this study are available on request from the corresponding author.

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
