# Peer review of "Hybrid Nanomaterial of Graphene Oxide Quantum Dots with Multi-Walled Carbon Nanotubes for Simultaneous Voltammetric Determination of Four DNA Bases"

_nanomaterials, 2023, doi:10.3390/nano13091509_

Round 1

Reviewer 1 Report

In this manuscript, the authors reported the synthesis of GOQD/MWCNT nanocomposites for the fabrication of electrochemical DNA base sensors. It was found that the GOQD/MWCNT nanocomposite-modified GCEs exhibited high performance for sensing four types of DNA bases. In addition, the fabricated electrochemical biosensors had practical application for detecting real saliva samples. It is an interesting work, although the synthesis nanocomposites are not so new. The manuscript is well-organized and good-written. All the conclusions are supported by the presented data. Therefore, this manuscript is recommended for publication at Nanomaterials after major revisions.

Special comments for the revision:

1.     In the title, “hybrid” and “nanocomposite” are duplicated, and “the” could be deleted.

2.     The authors have carried out some preliminary studies on this topic previously. Therefore, it is necessary for the authors to add more information on the novelty and significance of this current study.

3.     In the “Materials and methods” part, it is suggested for the authors to add a scheme to indicate clearly the synthesis process of the GOQD/MWCNT nanocomposites.

4.     It is necessary for the authors to add the AFM image of GOQDs, which will be helpful to see the height and size of the synthesized GOQDs.

5.     The characterization of the GOQD/MWCNT nanocomposites is weak. The authors should use spectral techniques to identify the formation of the GOQD/MWCNT nanocomposites.

6.     The authors did not carry out the analysis of the selectivity and stability of the fabricated electrochemical biosensors.

7.     The importance of the synthesized nanocomposites on the improvement of sensing is unknown. More discussion is needed.

Author Response

Please find it in the attached file.

Reviewer 2 Report

In the manuscript “Hybrid nanocomposite of graphene oxide quantum dots with multi-walled carbon nanotubes for the simultaneous voltammetric determination of four DNA bases”, the authors prepared a hybrid nanocomposite of GOQD-MWCNT using a solvothermal technique for the simultaneous voltammetric determination of four DNA bases. Although the topic is attractive in the fields of nanomaterials and biosensors, the authors should address some critical issues before I can recommend this article.

1.     The material is only characterized by XPS and TEM. It is quite common to investigate carbon-based nanomaterials with Raman spectroscopy.

2.     GOQD-MWCNT hybrid nanocomposite was prepared by mixing 10 mg of GOQDs and 50 mg MWCNTs 158 with 5 mL of ultra-pure water in a scintillation vial. Is the mixing ratio optimized? Did the authors test other mixing ratios?

3.     Figure 2, how do the authors assign the peaks with G, A, T, and C? The authors should cite the references.

4. In Figure 3, the authors are suggested the explain the peak shift depending on the pH values or cite the references.

5.     The stoichiometries of the chemical compounds should be subscript.

6.   The authors are strongly suggested to cite more up-to-date references. There are only two references published in 2021-2023.

Other minors:

1.     Line 145, “10oC” should be revised as “10oC”.

2.     Line 202, “surface are” should be revised as “surface area”.

3.   The resolutions of Figure 3 and Scheme 1 should be improved.

Author Response

Please find it in the attached file. 

Round 2

Reviewer 1 Report

In this revised version, the authors made great improvements according to the comments and suggestions of the referees. I am satisfied with these changes and therefore recommend the publication of this manuscript in current form.

Reviewer 2 Report

none